# Antioxidant and Anti-Inflammatory Activity of Coffee Brew Evaluated after Simulated Gastrointestinal Digestion

**DOI:** 10.3390/nu13124368

**Published:** 2021-12-05

**Authors:** Luigi Castaldo, Marianna Toriello, Raffaele Sessa, Luana Izzo, Sonia Lombardi, Alfonso Narváez, Alberto Ritieni, Michela Grosso

**Affiliations:** 1Department of Pharmacy, University of Naples “Federico II”, 49 Domenico Montesano Street, 80131 Naples, Italy; marianna.toriello@unina.it (M.T.); sonia.lombardi@unina.it (S.L.); alfonso.narvaezsimon@unina.it (A.N.); alberto.ritieni@unina.it (A.R.); 2Department of Molecular Medicine and Medical Biotechnology, School of Medicine, University of Naples “Federico II”, 5 Sergio Pansini Street, 80131 Naples, Italy; raffaele.sessa@unina.it (R.S.); michela.grosso@unina.it (M.G.); 3Health Education and Sustainable Development, University of Naples “Federico II”, 80131 Naples, Italy

**Keywords:** polyphenols, food-derived bioactive compounds, chlorogenic acids, inflammation

## Abstract

Coffee contains human health-related molecules, namely polyphenols that possess a wide range of pharmacological functions, and their intake is associated with reduced colon cancer risk. This study aimed to assess the changes in the anti-inflammatory and antioxidant activity of coffee after simulated gastrointestinal digestion. The evaluation of intracellular reactive oxygen species (ROS) levels in the HT-29 human colon cancer cell line and three in vitro spectrophotometric assays were performed to determine the antioxidant activity of the samples. Characterization of coffee composition was also assessed through a Q-Orbitrap high-resolution mass spectrometry analysis. The results highlighted that the levels of polyphenols in the digested coffee brews were higher than those of the non-digested ones. All assayed samples decreased the levels of intracellular ROS when compared to untreated cells, while digested coffee samples exhibited higher antioxidant capacity and total phenolic content than not-digested coffee samples. Digested coffee samples showed a higher reduction in interleukin-6 levels than the not-digested samples in lipopolysaccharide-stimulated HT-29 cells treated for 48 h and fewer cytotoxic effects in the MTT assay. Overall, our findings suggest that coffee may exert antioxidant and anti-inflammatory properties, and the digestion process may be able to release compounds with higher bioactivity.

## 1. Introduction

Coffee is recognized as one of the most popular beverages globally due to its appreciated taste and aroma as well as to its stimulating properties [1]. Although previous studies have reported adverse effects related to coffee consumption [2], mainly due to the presence of potential harmful contaminants such as acrylamide [3], a growing amount of scientific data support the potential benefits of regular coffee intake for human health [4,5]. Several epidemiological studies and meta-analyses confirmed an inverse relationship between regular coffee consumption and chronic diseases including cardiovascular disease, obesity, type 2 diabetes, Parkinson’s disease, and a wide number of cancer types [6,7,8,9]. Scientific studies indicate that coffee consumption may exert colorectal cancer protection [10], which could be due to the capacity of coffee constituents to inhibit the nuclear factor kappa B (NF-κB) pathway in colon cancer cells [11], suppress the activity of T-LAK cell-originated protein kinase (TOPK) [12], and inhibit neoplastic cell transformation [13] and colon cancer metastasis by attenuating phosphorylation of extracellular signal-regulated kinase (ERKs) [14]. Moreover, coffee consumption was also associated with reduced insulin resistance [15] and lower levels of inflammation [16], which plays a role in colorectal cancer risk [17,18]. The apparent beneficial properties of coffee consumption in colorectal cancer protection seem to be dose-dependent, as the risk decreased by 6% with each one cup of coffee per day, and the protection was about 30% for the highest coffee drinkers (up to 5 cups per day) [19]. Furthermore, in a recent meta-analysis involving 2,046,575 subjects, including 22,629 participants with colorectal cancer, high coffee consumption (≥5 cups per day) was found to be associated with a significant decreased risk of colorectal cancer [20]. On the other hand, Sinha et al. [21] showed that moderate consumption of decaffeinated coffee decreased the risk of colorectal cancer, suggesting that caffeine may not play a fundamental role to obtain such outcomes, and the protective effects have been related to the presence of a great concentration of bioactive molecules besides caffeine.

Coffee is known to contain a high amount of polyphenols and high molecular weight compounds, namely melanoidins, able to exert a number of positive effects on human health [22]. Melanoidins represent the newly produced compounds that appear in coffee during the last phase of the Maillard reaction [23]. The carbonyl groups of carbohydrate residues and the amino groups of proteins are involved in the formation of melanoidins [24]. As reported, the complex structure of coffee melanoidins includes low molecular weight active compounds, such as phenolics that are incorporated into them during the roasting process [25]. Chlorogenic acids (CGAs) are the most common polyphenolic compounds found in coffee [26,27]. Several studies have reported that CGAs possess a wide range of important physiological capacities such as free radical scavengers as well as anti-inflammatory and antioxidants properties, which may partly explain the disease-prevention benefit displayed by coffee consumption [28,29,30].

Nowadays, HT-29 human colon cancer cells, to study at the cellular level the effect of coffee on colon cancer cell growth, have been used in several experiments. Recently, spent coffee grounds have been reported to induce HT-29 cell apoptosis by reducing 8-iso-prostaglandin F2α and catalase after simulated colonic digestion [31]. Moreover, 5-caffeoylquinic acid (5-CQA) from coffee brews has been found to reduce ROS production in an HT-29 human colon cell model [32]. Furthermore, the level of roasting was reported to affect cell viability, and the results showed that lighter roasted coffee reduced HT-29 cell growth more than darker roasted samples [11].

To exert their biological activities, polyphenols need to be bioaccessible and bioavailable in the target tissue [33]. In this line, the colon stage seems to be crucial in polyphenols adsorption [34]. In fact, several in vitro and in vivo studies highlighted that about two-thirds of the ingested CGAs reach the lower intestine in their intact form, where they are metabolized by the gut microbiota [35,36,37], resulting in a wide range of chemical modifications and catabolic breakdowns that could account for their biological properties [38]. Moreover, scientific evidence has reported that, similarly to dietary fibers, the coffee melanoidins escape digestion to be eventually metabolized in the colonic stage by different microbial species, resulting in the hydrolytic release of antioxidants, mainly CGAs, linked to them [39].

Although the antioxidant and anti-inflammatory activities of coffee polyphenols have been widely studied, more in-depth knowledge is needed to clarify the effect of the gastrointestinal process on coffee composition and their bioactivities in protecting against colon cancer. Therefore, this study aimed to assess the changes occurring in polyphenol composition and the relative differences in antioxidant and anti-inflammatory properties of coffee brews after simulated gastrointestinal digestion (GiD) through the HT-29 human colon cancer cell model.

## 2. Materials and Methods

### 2.1. Chemicals and Reagents

Phenolic standards (purity >98%) were obtained as follows: 3-CQA, 3,4,-diCQA, gallic acid, caffeic acid *p*-coumaric acid, ferulic acid, quinic acid, and caffeine were acquired from Sigma-Aldrich (Saint Louis, MO, USA). Standards used for antioxidants tests were 2,3,5-triphenyltetrazolio chloride (TPTZ), 2,2′-azino-bis-3-ethylbenzthiazoline-6-sulphonic acid (ABTS), sodium sulfate (NaSO_4_), potassium persulphate (K_2_S_2_O_8_), 6-hydroxy-2,5,7,8-tetramethylchromane-2-carboxylic acid (commonly called Trolox), 1,1-diphenyl-2-picrylhydrazyl (DPPH), potassium thiocyanate (KCNS), calcium chloride dihydrate (CaCl_2_ 2 H_2_O), monosodium phosphate (NaH_2_PO_4_), anhydrous ferric chloride (FeCl_3_), sodium hydroxide (NaOH), potassium chloride (KCl), sodium bicarbonate (NaHCO_3_), and sodium acetate (C_2_H_3_NaO_3_) were provided from Sigma-Aldrich (Saint Louis, MO, USA).

The enzymes and standards used to simulate in vitro GiD were pepsin (≥2500 U/mg solid) from porcine gastric mucosa, α-amylase (1000–3000 U/mg solid) from human saliva, bile salt, pancreatin, Pronase E (bacterial protease from Streptomyces griseus, ≥3.5 U/mg solid), pancreatin (8 × USP) from porcine pancreas, and Viscozyme L were acquired from Sigma-Aldrich (Saint Louis, MO, USA).

Solvents, namely formic acid (FA), methanol (MeOH), hydrochloric acid (HCl), and water (UHPLC-MS grade), were acquired from Merk (Darmstadt, Germany). Deionized water (<18 MX cm resistivity) was acquired from Millipore (Bedford, MA, USA).

### 2.2. Sampling and Coffee Brew Preparation

Coffee bean (*Coffea arabica* L.) samples were acquired from a coffee shop located in Naples, Southern Italy. Coffee beans were roasted using a laboratory roaster (Probat-Sample roaster, Emmerich am Rhein, Germany) for 16 min at a temperature between 195 and 227 °C. The roasting degree reached a medium degree on the Agtrom disk of the SCA (tile 55). Then, roasted coffee beans were grounded to 0.4 mm particle size in a coffee grinder (DeLonghi, KG79, Treviso, Italy). The filtered coffee brews were prepared using an American machine (Aigostar Chocolate 30 HIK, Milan, Italy) working with a water temperature of 90 °C. Coffee brews were prepared from 50 g of coffee powder and a volume of 600 mL of hot deionized water as declared by the manufacturer. The coffee brews were collected, freeze-dried, and stored until the analysis.

### 2.3. UHPLC and Orbitrap HRMS Analysis

Chromatographic analysis was carried out through an UHPLC (Dionex UltiMate 3000, Thermo Fisher Scientific, Waltham, MA, USA) prepared with a Quaternary UHPLC pump, a degassing system, and an autosampler device. Separation of polyphenols and caffeine was performed with a thermostated Kinetex (25 °C) column F5 (50 × 2.1 mm, 1.7 µm particle size, Phenomenex, Torrance, USA).

The eluent phase consisted of H_2_O (A) and MeOH (B) both prepared at 0.1% of FA. The separation gradient started with 100% A for 1 min, decreased to 20% A in 2 min, and decreased again reaching 0% A in 3 min. Then, the gradient rose up again reaching 100% A in 2 min and was held for 2 min for column re-equilibration. The injection volume was 5 µL, whereas 0.5 mL/min was the flow rate.

Mass spectrometry analysis was performed through a Q-Exactive Orbitrap mass spectrometer (Thermo Fischer Scientific, Waltham, MA, USA). An electrospray (ESI) source was simultaneously operated in fast positive/negative ion switching mode, setting two scan events (Full ion MS and All ion fragmentation: AIF) for all investigated compounds.

Full MS/AIF experiments were performed with the settings: automatic gain control (AGC) target, 1e6; mass resolution, 35,000 full width at half maximum (FWHM); microscans, 1; maximum injection time, 200 ms; and scan range, 80–1200 *m*/*z* for full MS analysis. AIF mode conditions were: maximum injection time 200 ms; mass resolving power to 17,500 FWHM, AGC target, 1e6; scan time, 0.10 s; scan range, 80–1200 *m*/*z*; isolation window, 5 *m*/*z*; and retention time to 30 s. The collision energies were varied in the range of 10–60 eV. In both scan events the instrument was set to auxiliary gas, 10; spray voltage, 3.5 kV; sheath gas 45; and capillary temperature to 275 °C.

Identification was performed considering exact mass measurements at a mass tolerance of 5 ppm. Data treatment was carried out using Quan/Qual Browser Xcalibur software 3.1.66.19 (Xcalibur, Thermo Fischer Scientific, Waltham, MA, USA).

### 2.4. Simulated GiD

The in vitro digestion process was carried out according to the protocol proposed from INFOGEST network [40]. The simulated fluids (salivary: SSF; gastric: SGF; and intestinal: SIF) were prepared according to the proportion of salts described by Izzo et al. [41] (Appendix A).

In short, to simulate the oral phase, 5 mL of brewed coffee brew was mixed with 975 µL of water, 3.5 mL of SSF, 25 µL of 0.3 M CaCl_2_ (H_2_O)_2_, and 0.5 mL of α-amylase enzyme (50 mg of 250 U/mg solid). Then, the solution was incubated in a shaker bath for 2 min at 37 °C. Afterward, 7.5 mL of SGF, 695 µL of water, 1.6 mL of pepsin solution (2000 U/mL), and 5 µL of 0.3 M CaCl_2_ (H_2_O)_2_ were added to the mixture to simulate gastric conditions. The pH of the solution was reduced to 3 with 1 M HCl, and after that, the sample was incubated for 2 h at 37 °C. Finally, to simulate the intestinal conditions, 2.5 mL of bile salt solution (65 mg/mL), 11 mL of SIF, 1.3 mL of water, 5 mL pancreatin solution (100 U/mL of trypsin activity), and 40 µL of 0.3 M CaCl_2_ (H_2_O)_2_ were added. The pH value of the mixture was increased to 7 using NaOH 1 M. After that, the solution was incubated for 120 min at 37 °C. Then, the samples were centrifuged at 5000× *g* at 37 °C for 10 min, and the remaining pellets were collected. In order to simulate the colonic stage, the pellets were treated according to a previously described procedure [42]. Briefly, 5 mL of Pronase E solution (1 mg/mL) was added. The mixture was incubated for 1 h at 37 °C. Finally, 5 mL of water and 150 µL of Viscozyme L were added to the mixture and incubated for 16 h at 37 °C. After that, the samples were centrifuged at 5000× *g* for 10 min. The supernatants were collected, freeze-dried, and stored until the analysis.

### 2.5. In Vitro Antioxidant Activity

The antioxidant activity of the digested and not-digested coffee brew samples was assessed by using two different free radical scavenging activity methods, including ABTS and DPPH tests, and the ferric ion reducing antioxidant power assay, namely FRAP. Data obtained were expressed as mmol of Trolox equivalents (TE) per gram of sample dry weight (DW). Results were calculated from the calibration curve prepared in triplicate at 6 concentration levels (5–200 µM of Trolox).

#### 2.5.1. FRAP Assay

The FRAP test was carried out according to a methodology previously described [43]. In short, the FRAP reagent was prepared by mixing 12.5 mL of acetate buffer (0.3 M, pH 3.6), 1.25 mL of FeCl_3_ solution (20 mM), and 1.25 mL of TPTZ solution (10 mM). Then, 150 µL of sample was added to 2.85 mL of FRAP reagent. After 4 min, the values of absorbance at 593 nm were immediately recorded.

#### 2.5.2. ABTS Assay

The ABTS test was carried out based on the procedure described by Dini et al. [44]. In short, forty-four microliters of K_2_S_2_O_8_ (2.5 mM) was added to aqueous ABTS (2.5 mL; 7 mM). After 16 h at room temperature, the solution was diluted with EtOH to reach an absorbance value of 0.70 (±0.02) at 734 nm. Then, 100 µL of sample was added to 1000 µL of ABTS radical working solution. After 3 min, the values of absorbance were immediately recorded.

#### 2.5.3. DPPH Assay

The DPPH test was performed using the procedure suggested by Dini et al. [45]. Briefly, 4 mg of DPPH was diluted with MeOH until the absorbance value reached 0.9 (±0.02) at 517 nm. Afterward, 1 mL of DPPH radical working solution was added to 0.20 mL of sample. After 10 min, the values of absorbance were immediately recorded.

### 2.6. Total Phenolic Content Assay

The Folin–Ciocalteu method was performed to evaluate the total phenolic content (TPC) value in accordance with the procedure previously described [46,47]. In brief, 0.125 mL of Folin–Ciocalteu reagent and 0.50 mL of deionized H_2_O were mixed with 0.125 mL of sample. After 6 min of incubation at room temperature, 1 mL of H_2_O and 1.25 mL Na_2_CO_3_ solution (7.5%) were added to the solution. Finally, after 90 min of incubation, the values of absorbance at 760 nm were immediately recorded. Results were calculated from the calibration curves prepared in triplicate at 6 concentration levels (0.25–0.01 mg/mL of gallic acid).

### 2.7. Cell Culture

The HT-29 human colon carcinoma cell line was obtained from ATCC (American Type Culture Collection, Manassas, VA, USA). Cells were kept in Dulbecco’s Modified Eagle’s Medium, high glucose, in the presence of 10% fetal bovine serum, 4 mM glutamine, streptomycin (10 mg/mL), and penicillin (10 U/mL) as previously reported [48]. The cell culture was maintained at 37 °C and humidified atmosphere (5% CO_2_). Moreover, in order to maintain growth in the log phase, cells were subcultured by trypsinization (all reagents from Sigma-Aldrich, Saint Louis, MO, USA). Cells have been routinely checked for mycoplasma contamination with the PCR Mycoplasma Test Kit (AppliChem A3744, Darmstadt, Germany). Only cells negative for mycoplasma contamination were used.

### 2.8. Cell Treatment

Individual stock solution of assayed samples (digested and not-digested coffee brews) at 2 mg/mL were prepared in cell culture media. Therefore, in order to set the optimal experimental conditions to evaluate intracellular ROS levels and to perform cell viability assays and Western blot analysis, HT-29 cells were initially treated with different concentrations of the assayed samples (0.250 to 2 mg/mL) as previously reported [11] for 24, 48, and 72 h.

### 2.9. Analysis of Cell Viability

Cell viability was evaluated through spectrophotometric assay using the thiazolyl blue tetrazolium bromide (MTT) test [49,50]. Briefly, 24 h before treatment, cells were plated into 96-well plates (100 µL cell suspension per well) at a density of 5.5 × 10^5^ cells/mL. Then, cells were treated with the investigated samples. After 24, 48, and 72 h of treatment, 10 µL of the MTT labeling reagent provided by the Cell Proliferation Kit I (Roche, Mannhein, Germany) was added to each well. After 4 h, 100 µL of detergent solubilization buffer 1× (10% SDS in 0.01M HCl) was added to dissolve the insoluble purple formazan products into a colored solution according to the procedure recommended by the manufacturer. Measurement of the soluble formazan product in each well was carried out by photometric reading at 570/690 nm on a Synergy H1 Hybrid Multi-Mode Microplate Reader (BioTek, Winooski, VT, USA).

### 2.10. Evaluation of Intracellular ROS Level

Intracellular ROS level production was estimated using the fluorescent dye 2′7′-dichlorodihydrofluorescein diacetate (H_2_DCF-DA) [51]. Cells were plated in 96-well black plates (100 µL cell suspension per well) at a density of 5 × 10^5^ cells/mL. After 48 h treatment with different concentrations of the assayed samples (0.250 and 0.500 mg/mL), the cells were washed twice with Dulbecco’s Phosphate Buffered Saline (DPBS). In order to test ROS levels under challenging conditions, after 24 h treatment with 0.250 mg/mL of assayed samples, cells were exposed to 10 ng/mL LPS for an additional 24 h to induce ROS production [52,53,54]. All cell samples were incubated and labelled with 10 µM of H_2_DCF-DA diluted in Hank’s Balanced Salt Solution (HBSS) at 37 °C for 20 min in the dark. Then, the dye was removed, and the cells were washed twice with PBS. Cells treated with 100 µM hydrogen peroxide (H_2_O_2_) were used as positive control. Therefore, fluorescence was measured on a Synergy H1 Hybrid Multi-Mode Microplate Reader (BioTek) at excitation/emission wavelengths of 485/538 nm.

### 2.11. Protein Extraction and Western Blot Analysis

HT-29 cells were plated into 24-well plates at a density of 1.4 × 10^5^ cells/mL. Cells were treated with the assayed samples at the concentrations of 0.250 and 0.500 mg/mL. After 24 h, cells were exposed further for 24 h to lipopolysaccharide (LPS) to induce inflammation (10 ng/mL). For protein extraction, cells were collected and washed twice with 4 mL of PBS by centrifugation at 1000 rpm for 10 min at 4 °C. The pellets were resuspended with 50 µL of RIPA Buffer (Thermo Fisher Scientific, Waltham, MA, USA), 0.5 µL of the protein inhibitor cocktail mixture (Sigma-Aldrich), and incubated for 30 min on ice. Then, the whole cell lysates were collected [55,56,57]. Protein concentration was measured by spectrophotometric analysis, according to the Bradford method [58] with the Bio-Rad protein analysis reagent (Bio-Rad Laboratories, Hercules, CA, USA). Western blot analysis was performed on 20 µg of total protein extracts. Cell extracts were separated using 4–15% Mini-Protean TGX Stain-Free Precast Gels reagent (Bio-Rad Laboratories) and transferred to membranes using Trans-Blot Turbo Transfer System (Bio-Rad Laboratories). Antibodies against actin (1:1000 dilution/#SC-1616; Santa Cruz Biotechnology, Santa Cruz, CA, USA), IL-6 (1:1000 dilution/#ab9324; Abcam, Cambridge, UK), IL-10 (1:1000 dilution/#sc-1783; Santa Cruz Biotechnology), and NF-kB p65 subunit (1:2000 dilution/#ab1604a; Millipore) were diluted in TBT-Tween 20 (TBT-T) buffer containing 5% of milk and applied to membranes, followed by overnight incubation at 4 °C. The next day, filters were washed three times with TBT-T for 5 min and incubated for 45 min with respective secondary antibodies conjugated to peroxidase. The antigen–antibody complexes were then detected using Clarity Western ECL Substrate (Bio-Rad Laboratories). Quantitative densitometry of bands was carried out by analyzing ChemiDoc system (Bio-Rad Laboratories), and the quantification of the signal was performed by ImageJ as previously reported [59,60].

### 2.12. Statistical Analysis

Tukey’s test, at the level of significance *p*-value ≤ 0.05, was used to evaluate differences in antioxidant activity and TPC content between average values of the assayed samples. Statistical differences between untreated control and treated cells were calculated using the two-way analysis of variance (ANOVA) or one-way ANOVA test followed by Dunnett’s multiple comparisons test and/or multiple Student’s t-tests, where appropriate. The level of significance was set at *p*-value ≤ 0.05, *p*-value ≤ 0.01 to be significant (* and **, respectively), and at *p*-value ≤ 0.001 to be highly significant (***). All analyses were performed in triplicate. Data treatment was performed using Stata 12 software (StataCorp LP, College Station, TX, USA).

## 3. Results

### 3.1. Identification of Polyphenol Compounds and Caffeine in the Assayed Samples Using UHPLC-Q-Exactive Orbitrap

UHPLC-Q-Orbitrap HRMS analysis was conducted to identify active molecules (*n* = 16), including caffeine, CGAs, and hydroxycinnamic acids, in the digested and not-digested samples. Optimal separation of the studied compounds in the assayed samples was achieved by the UHPLC system in a total run time of 13 min. However, the 5- and 4-feruloylquinic acids (FQAs) isomers were quantified as a sum, due to the suboptimal separation. The identification of active compounds was assessed in both Full ion MS and AIF modes. All experiments were conducted in negative ESI^−^ mode, except for caffeine which showed improved fragmentation patterns in positive ESI^+^ mode. Mass parameters including chemical formula, ion assignment, measured and theoretical mass (*m*/*z*), retention time (RT), and accuracy are shown in Table 1. Isomer identification of *p*-coumaroylquinic acid (*p*-CoQA, *m*/*z* 337.09289), dicaffeoylquinic acid (diCQA, *m*/*z* 515.11950), feruloyl-caffeoylquinic and caffeoyl-feruloylquinic acids (FCQA and CFQA, *m*/*z* 529.13245), and CQA (*m*/*z* 353.08780) was performed by comparing the RT of the peaks with those of the standards and by comparison of fragmentation pattern obtained with data previously reported in the literature.

### 3.2. Quantification of Polyphenol Compounds and Caffeine in the Assayed Samples Using UHPLC-Q-Exactive Orbitrap

Brewed coffee samples (digested and not-digested) were investigated using a UHPLC-Q-Orbitrap HRMS method. The quantitative analysis of the target compounds was carried out using calibration curves, and regression coefficients >0.990 were obtained for all analytes. Some important CGAs (*n* = 11), phenolic acids (*n* = 4), and caffeine were found in both digested and not-digested samples. 

Total CGAs were quantified at concentrations ranging from 68.92 (not-digested samples) up to 81.50 mg/g (digested samples), as shown in Table 2. In the here investigated samples, CQAs were found as the most detected compounds ranging from 43.31% (not-digested samples) to 46.45% (digested samples) of total CGAs. The CQAs were revealed in a concentration range between 29.85 and 37.86 mg/g. Notably, 5-CQA showed to be the more relevant CGA, being quantified in digested and not-digested samples at a mean value of 25.97 and 21.40 mg/g, respectively. Regarding the FQAs isomers, 3- and 4-FQA represented from 30.48 to 30.64% of total CGAs detected in the investigated samples, in a concentration range between 21.11 and 24.84 mg/g. As far as *p*-CoQAs are concerned, 5-pCoQA was the most abundant compound detected in both digested and not-digested samples at a mean value of 7.53 and 7.19 mg/g, respectively. Moreover, diCQAs mainly represented by 3,4- and 3,5-diCQA were detected at a concentration level of 2.61 and 2.23 mg/g in digested and not-digested samples, respectively. Concerning 3,4-FCQA and 4,5-CFQA isomers, they were found as the less relevant compounds being measured at concentrations ranging from 0.83 up to 1.41 mg/g.

On the other hand, some important phenolic acids were quantified in the here-assayed samples. Digested samples showed higher total phenolic content (9.57 mg/g) when compared to not-digested samples (5.29 mg/g). Moreover, ferulic acid was found at a concentration level significantly higher than the other assayed phenolic acids, ranging from 2.71 (not-digested samples) up to 4.12 mg/g (digested samples). Apart from those, caffeine was assessed in the herein investigated samples. Not-digested samples showed higher caffeine content (19.96 mg/g) than the digested samples (17.68 mg/g), as shown in Table 2.

### 3.3. Antioxidant Activity and Total Phenolic Content

The antioxidant activity of the digested and not-digested samples was measured and compared using three different tests (FRAP, DPPH, and ABTS). Data were displayed as millimoles of Trolox per 100 g of coffee brew dried matter (mean value and ±SD). The results are reported in Table 3. In all assayed methods, the digested coffee brews showed significantly higher antioxidant capacity (*p*-value ≤ 0.05) than the not-digested coffee brews. Furthermore, the results revealed that the digested samples showed a two-fold increase in antioxidant activity in both DPPH and ABTS tests (51.4 vs. 22.8 and 89.2 vs. 44.6 mmol Trolox/100 g DW, respectively). As shown in Table 3, the digested samples also showed a significantly higher TPC value (*p*-value ≤ 0.05) compared to not-digested coffee brew samples. In addition, the blank control resulting from the in vitro digestion experiments was tested, and the results are presented in Appendix A. Comparing the FRAP, DPPH, and ABTS data against TPC values measured through the Folin–Ciocalteu test, strong positive correlations were observed (R^2^ = 0.892, 0.984, and 0.892 for TPC vs. FRAP, vs. DPPH, and vs. ABTS, respectively).

### 3.4. Effect of not-Digested or Digested Coffee on Cell Viability in HT-29 Cells

MTT assay was performed to evaluate the potential cytotoxicity exerted by not-digested and digested coffee extracts in HT-29 cells at different concentrations according to literature data [11]. As shown in Figure 1, at concentrations ranging from 0.250 mg/mL to 0.750 mg/mL, both extracts showed similar low effects on cell viability, although it has to be noted that, at the lowest concentration used in this study (0.250 mg/mL), an increase in cell viability was observed in both cases. Conversely, at higher concentrations (1–2 mg/mL) cell viability was significantly reduced for both extracts as compared to untreated control. In this context, it is interesting to note that not-digested coffee samples showed dramatic reduction in cell viability after treatment with 2 mg/mL for 24 h (14%), 48 h (10%), and 72 h (3%). In contrast, exposure to digested samples at the same concentration resulted in a less marked reduction in cell viability, with 77% of viable cells detected at 24 h, 65% at 48 h, and 33% at 72 h (Figure 1A–C), thus suggesting that digested coffee exhibits less cytotoxic effects on HT-29 cells than not-digested coffee samples. To exclude effects mediated by the digestion fluid, the blank control resulting from in vitro digestion was analyzed by MTT assay and compared with untreated cells. Results (Figure 1D) indicated no significant differences between the blank control and untreated cells at 24 h, 48 h, and 72 h of treatment.

### 3.5. Evaluation of Intracellular ROS Level

Changes in intracellular ROS levels in HT-29 cells were evaluated using the H_2_DCF-DA assay after 24 h, 48 h, and 72 h exposure to non-toxic doses (0.250 and 0.500 mg/mL) of non-digested and digested coffee samples. Cells treated with 100 µM H_2_O_2_ were used as positive control. The data summarized in Figure 2 show that treatment with not-digested or digested coffee samples decreased the levels of intracellular ROS levels as compared to untreated cells. Even in this case, different trends of antioxidant activity were detected in not-digested and digested coffee samples, with digested coffee samples showing a more pronounced antioxidant capacity, more evident at the longer exposure times (48 h and 72 h) (Figure 2B,C). In addition, to verify the real effects mediated by the digested coffee and to exclude effects mediated by the digestion fluid, a blank control resulting from in vitro digestion was included in the H_2_DCF-DA assay showing no significant differences with untreated cells at each exposure time.

### 3.6. Anti-Inflammatory Effects of Not-Digested and Digested Coffee

Western blotting analysis was performed to determine the antioxidant effects of the assayed samples in modulating the expression of the pro-inflammatory protein IL-6 and NF-kB p65 subunit and the expression of the anti-inflammatory protein IL-10. Data obtained from HT-29 cells treated for 24 h with digested and not-digested coffee and then co-exposed to LPS for additional 24 h as pro-inflammatory action are shown in Figure 3. Notably, the results highlighted that the digested coffee samples showed lower expression levels of IL-6 and NF-kB p65 subunit levels than the not-digested samples (1.35 vs. 0.90 arbitrary units and 1.08 vs. 0.90 arbitrary units, respectively), indicating that the digestion process releases molecular compounds with more effective anti-inflammatory activity than the native coffee samples (Figure 3A,B,D,E). Furthermore, to underline the anti-inflammatory effect of the assayed digested coffee sample, expression levels of the anti-inflammatory cytokine IL-10 were examined, resulting to be more enhanced in response to the treatment with digested coffee compared to not-digested coffee samples (Figure 3A,C). To better investigate the antioxidant capacity of not-digested and digested coffee, we also analyzed intracellular ROS levels using the H_2_DCF-DA assay after 48 h under challenging conditions after LPS treatment. In fact, according to scientific evidence, LPS treatment promotes intestinal inflammation by inducing intracellular ROS production and oxidative injury in HT-29 cells [52,53,54]. Based on these observations, LPS-treated cells, similarly to H_2_O_2_, showed 30% increased ROS levels compared with the untreated control, thus representing a challenging condition for oxidative stress. After stimulation with LPS, treatments with 0.25 mg/mL of not-digested or digested coffee significantly reduced the levels of ROS in HT-29 cells relative to LPS-stimulated cells by 15% and 30%, respectively. These results confirmed the pro-oxidant status induced by LPS stimulus and suggest that the more protective effects observed for digested coffee treatment are mediated, at least in part, by antioxidant mechanisms of ROS scavenging (Figure 4).

## 4. Discussion

The present study aimed to provide useful information regarding the bioactivities of coffee brew after simulated GiD. Despite the many scientific works present in the literature concerning the beneficial activities of coffee polyphenols in protecting against colon cancer, the study of bioactivities of compounds released after in vitro digestion has been barely investigated to date. Hence, the main goal of this work was to investigate the antioxidant and anti-inflammatory activities of coffee brews after GiD through the HT-29 human colon cancer cell model. The INFOGEST protocol, recognized as one of the most suitable procedures capable of simulating the physiological digestion process, was performed until the duodenal phase, whereas the associated action of Pronase E (mix of bacterial protease) and Viscozyme L (mix of carbohydrases) was used to mimic the gut microbiota activity since the INFOGEST protocol does not cover the large intestinal phase. Although the use of the fecal inoculum is recognized as the most suitable method to simulate in vitro colonic digestion, an ever-expanding amount of scientific works propose as an effective alternative to reproduce the intestinal fermentation the use of a mix of bacterial enzymes, such as Viscozyme L and Pronase E [61,62,63,64].

Moreover, to better understand the biological activity of coffee, changes in active molecules were also evaluated using a Q-Exactive Orbitrap mass spectrometer. In detail, four phenolic acids, eleven CGAs, and caffeine were identified, quantified, and compared in digested and not-digested coffee brew samples.

Overall, the obtained data highlighted that the filtered coffee brew may represent a rich source of bioactive molecules and the GiD process could affect the monitored active compounds present in a cup of filtered coffee. Moreover, the results clearly indicate that 5-CQA was the predominant active compound found in digested and not-digested coffee samples. Our findings are comparable to those of Farah et al. [65] who reported that 5-CQA accounted for 41 to 48% of total CGAs found in different brewed coffee. Concerning total CGAs occurrence in assayed samples, the levels found in the digested coffee brews were higher than the non-digested ones. These data were in accordance with previously published evidence showing increased CGAs content after the colonic phase. Bekerdam et al. [66] demonstrated that enzymatic hydrolysis using bacterial proteases to mimic the colon stage was able to release CGA and phenolics from coffee samples.

As reported by Pérez-Burillo et al. [67], melanoidins, which possess significant antioxidant potential, contain in their complex structure a wide range of antioxidant molecules, mainly represented by CGAs and phenolic acids including hydroxycinnamic acids such as ferulic, caffeic, and *p*-coumaric acids [68].

Recently, several scientific studies reported that the coffee melanoidins escape the upper digestion process; however, the activity of gut microbiota could play a role in the release of dietary polyphenols, including CGAs, incorporated into melanoidins. In the present work, the amount of hydroxycinnamic acids found in digested samples was higher than in the not-digested ones. This is in agreement with the literature [69], suggesting that low-molecular catabolites such as the phenolic acids mentioned above represent the main CGA breakdown metabolites mediated by gut microbiota.

CGAs are well recognized as potent molecules with antioxidant and anti-inflammatory activity [70]. Strong scientific evidence has highlighted the ability of CGAs to delay glucose absorption, modulate glucose and lipid metabolism, thus helping to prevent degenerative and non-degenerative diseases [28,71,72,73].

On the other hand, antioxidant capacity and TPC values of digested and not-digested samples were also assessed and compared. The obtained data showed that both the TPC values and antioxidant capacity of coffee significantly increased after the simulated gastrointestinal process. Regarding the antioxidant activity, the digested samples showed a two-fold increase, in both DPPH and ABTS tests, compared with the not-digested samples, whereas the TPC value of the digested samples increased about 10% after the colonic stage. Similarly, such phenomena were also observed in our previous study [34], which showed that after the simulated digestion process, the antioxidant capacity and TPC value increased in all types of coffee brews assayed, namely espresso, americano, and instant coffee. Moreover, similar data were obtained by Campos-Vega et al. [74], who demonstrated that both antioxidant capacity and the colon bioaccessibility of polyphenols from spent coffee, rich in CGAs and melanoidins, were significantly higher after colonic fermentation compared to the non-digested samples. Interestingly, results obtained from FRAP, DPPH, and ABTS assays were positively correlated with TPC values, highlighting that these tests could be suitable to provide reliable information about the antioxidant molecules released after the colonic stage.

The antioxidant capacity of the assayed samples was also estimated by assessing changes in intracellular ROS levels in HT-29 cells. In accordance with literature that support a protective effect of coffee bioactive molecules against oxidative stress, our findings showed that coffee has an important effect in preventing ROS production, which is partly due to its antioxidant activity. It is interesting to note that both digested and not-digested samples induced a significant decrease in the levels of intracellular ROS. In particular, when HT-29 human colon cancer cells were treated for 48h with the assayed samples, the digested coffee highlighted more effective antioxidant activity than the not-digested ones, suggesting that the higher number of phytochemicals found in the samples after the digestion process could play an important role in managing ROS production.

On the other hand, no cytotoxicity effects were revealed in HT-29 cells analysis for both not-digested and digested coffee samples tested at a concentration ranging between 0.250 to 0.750 mg/mL. However, at the highest concentrations tested (1 to 2 mg/mL), not-digested coffee samples showed a drastic reduction in cell viability. Nevertheless, the results showed that the gastrointestinal process was able to release compounds that exhibited a less marked reduction in cell viability than not-digested coffee samples. As a whole, our results are in agreement with several previous studies. In fact, Mojica et al. [11] compared the impact of the different coffee extracts on the growth inhibitory activity of HT-29 cells. The cells were treated with 1 to 50× dilutions of the coffee stock solutions, and the coffee samples were tested without a previous simulated GiD. The authors reported that polyphenolic compounds in different coffee extracts were able to reduce cell growth, revealing that bioactive phytochemicals could positively influence cell survival, avoiding changes in metabolism and mitochondrial structure. Furthermore, Choi et al. [75] reported that among the bioactive compounds present in coffee, including kahweol, caffeic acid, CGA, and caffeine, only kahweol exhibited significant cytotoxicity in HT-29 cell lines, increasing the expression of caspase-3 and inhibiting the HT-29 cell growth. A further in vitro study [76] on the effect of coffee CGA on HT-29 cells reported that treatment with 1 mM CGA significantly reduced the growth rate of the HT-29 cell by 52%.

LPS, an endotoxin of the cell wall of Gram-negative bacteria, is commonly used as an inflammatory stimulus to induce the production of cytokines [52,57]. Our data showed reduced protein levels of the pro-inflammatory IL-6 and NF-kB in HT-29 cells treated for 48h with coffee extracts and then exposed to LPS. The presence of CGAs in coffee appears to be fundamental. In fact, previous studies [28,65,77] have highlighted that coffee CGAs in LPS-stimulated Caco-2 cells were able to downregulate the key transcription factors including the pro-inflammatory cytokines IL-6 and tumor necrosis factor-alpha. In addition, our findings reveled increased levels of IL-10, an anti-inflammatory cytokine, after the treatment with digested coffee than with not-digested coffee extract. LPS treatment also is involved in the intestinal inflammation by inducing intracellular ROS production and oxidative injury in several cell models, including HT-29 cells [52,53,54].

Therefore, we observed that LPS treatment, representing a challenging condition for oxidative stress, induced increased oxidative stress levels in HT-29 cells. Pretreatment with both coffee samples effectively inhibited LPS-induced ROS levels, more efficiently with the digested coffee extract, suggesting that the more effective anti-inflammatory effects observed for this coffee treatment are mediated, at least in part, by antioxidant mechanisms of ROS scavenging.

## 5. Conclusions

In conclusion, coffee was demonstrated to exert anti-inflammatory and antioxidant properties. In fact, coffee samples inhibited intracellular ROS and pro-inflammatory pathways, thus exerting potential protective effects against cancer transformation on human colon cells. Therefore, the use of simulated GiD could represent a novel and useful strategy to better evaluate the effects of bioactive compounds present in a food matrix on the gut, chronically exposed to ROS injury and inflammation stimuli, where these compounds can exert their beneficial health effects. However, further in vivo studies are needed to assess the antioxidant and anti-inflammatory potential of coffee extracts and their molecular mechanism.

## Figures and Tables

**Figure 1 nutrients-13-04368-f001:**
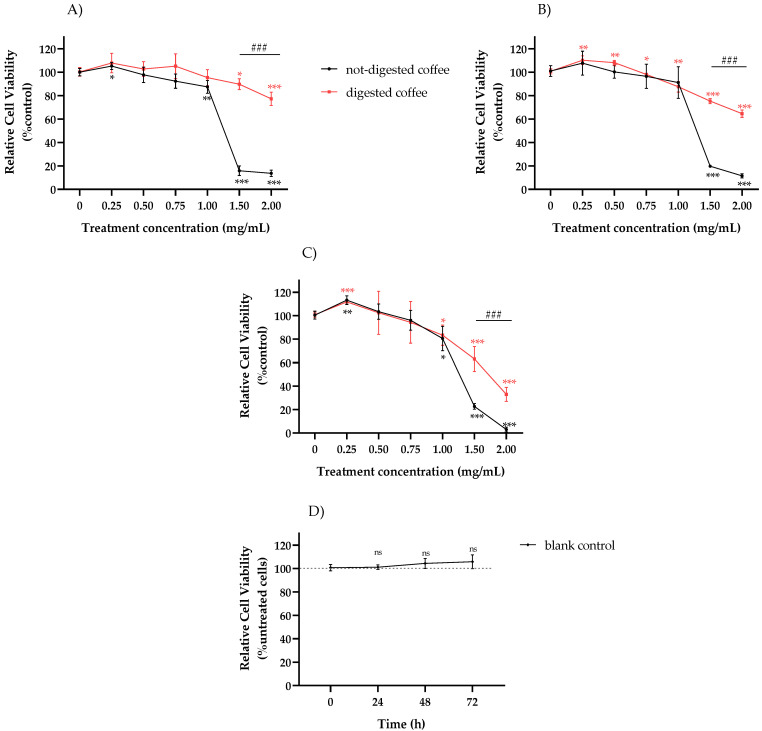
Evaluation of response to digested or not-digested coffee samples in HT-29 cells. The effect of treatment with not-digested or digested coffee extract at 0.25, 0.50, 0.75, 1.00, 1.50, and 2.00 mg/mL on cell viability was estimated by MTT assay after 24 h (**A**), 48 h (**B**), and 72 h (**C**) as compared to untreated control. The effect of treatment with vehicle (blank control) resulting from in vitro digestion on cell viability was evaluated by MTT assay after 24 h, 48 h, and 72 h (**D**), as compared to untreated cells. No significant differences were observed between blank control and untreated cells. ns: not statistically significant. Mean ± SD of three independent experiments were plotted on the graph. Differences were considered significant when *p*-value ≤ 0.05 and *p*-value ≤ 0.01 and highly significant when *p*-value ≤ 0.001. * *p* ≤ 0.05, ** *p* ≤ 0.01, and *** *p* ≤ 0.001 versus untreated control (calculated as fold-change relative to untreated cells, arbitrarily set at 100%); ^###^ *p* ≤ 0.001 not-digested coffee versus digested coffee.

**Figure 2 nutrients-13-04368-f002:**
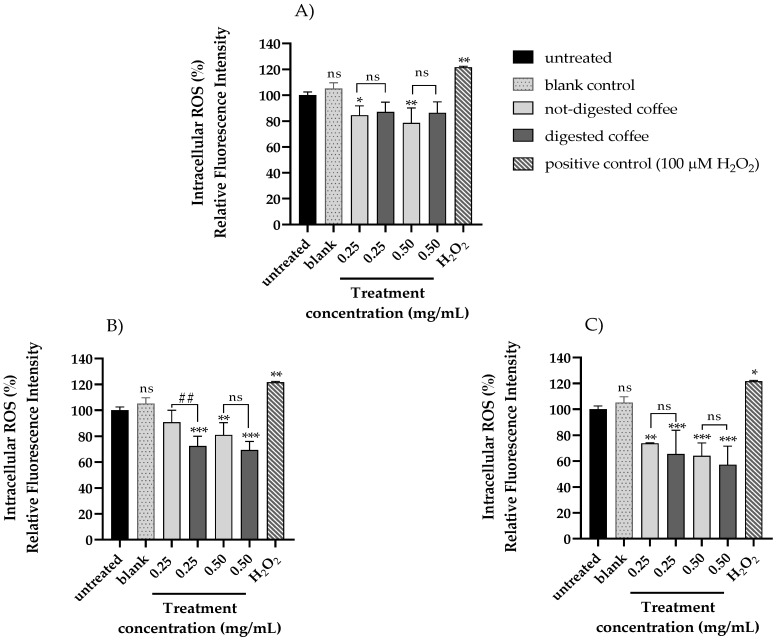
Evaluation of intracellular ROS level in HT-29 cells treated with not-digested or digested coffee samples. Intracellular ROS level was assessed by H_2_DCF-DA assay on HT-29 cells treated for 24 h (**A**), 48 h (**B**), and 72 h (**C**) with different concentrations of digested and not-digested coffee samples (0.25 and 0.5 mg/mL) and with vehicle (blank control) resulting from in vitro digestion related to untreated cells. Cells treated with 100 µM H_2_O_2_ were used as positive control. Mean ± SD of three independent experiments were plotted on the graph. Differences were considered significant when *p*-value ≤ 0.05 and *p*-value ≤ 0.01 and highly significant when *p*-value ≤ 0.001. * *p* ≤ 0.05, ** *p* ≤ 0.01, and *** *p* ≤ 0.001 versus untreated control (calculated as fold-change relative to untreated cells, arbitrarily set at 100%); ^##^ *p* ≤ 0.01 not-digested coffee versus digested coffee. No significant differences were observed between blank control and untreated cells at 24 h, 48 h, and 72 h. ns: not statistically significant.

**Figure 3 nutrients-13-04368-f003:**
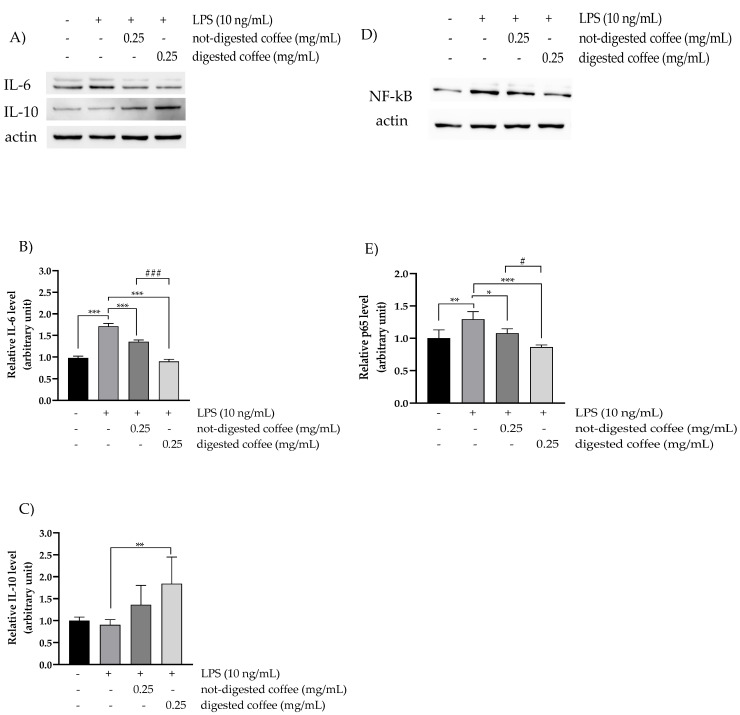
Anti-inflammatory effects of digested coffee samples in LPS-stimulated HT-29 cells. Western blot analysis of the expression levels of interleukin-6 (IL-6) (**A**), interleukin-10 (IL-10) (**A**), and NF-kB (p65 subunit) (**D**) in total cell lysates from untreated control and LPS-stimulated HT-29 cells treated for 48 h with not-digested coffee or digested coffee samples loading. (**B**,**C**,**E**) Densitometric analysis of Western blots. Band intensities were quantified and normalized to actin used as control. All data were analyzed for statistical significance by two-way ANOVA, followed by Dunnett’s multiple comparison test where appropriate. Differences were considered significant when *p*-value ≤ 0.05 and *p*-value ≤ 0.01 and highly significant when *p*-value ≤ 0.001. * *p* ≤ 0.05, ** *p* ≤ 0.01, and *** *p* ≤ 0.001 versus untreated control (calculated as fold-change relative to untreated cells, arbitrarily set at 1); ^#^ *p* ≤ 0.05, and ^###^ *p* ≤ 0.001 not-digested coffee versus digested coffee. Differences were considered significant when *p*-value ≤ 0.05 and highly significant when *p*-value ≤ 0.0001. * *p*-value ≤ 0.05, ** *p*-value ≤ 0.0001 versus LPS-stimulated control; ^#^ *p*-value ≤ 0.05, ^###^ *p* ≤ 0.001 not-digested coffee versus digested coffee.

**Figure 4 nutrients-13-04368-f004:**
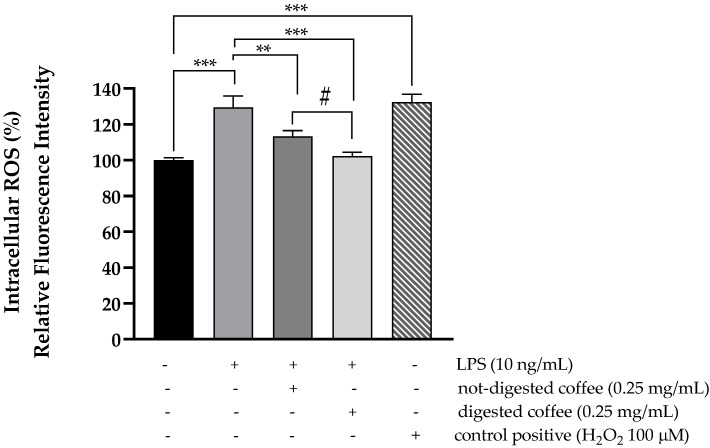
Evaluation of intracellular ROS level in HT-29 cells treated with not-digested and digested coffee after LPS treatments. Intracellular ROS levels were assessed by H_2_DCF-DA assay in LPS-stimulated HT-29 cells treated for 48h with digested and not-digested coffee samples (0.25 mg/mL) compared with untreated cells. H_2_O_2_ (100 µM) was used as positive control. Mean ± SD of three independent experiments were plotted on the graph. Differences were considered significant when *p*-value ≤ 0.05, *p*-value ≤ 0.01 and highly significant when *p*-value ≤ 0.001. ** *p* ≤ 0.01, and *** *p* ≤ 0.001 versus untreated control (calculated as fold-change relative to untreated cells, arbitrarily set at 100%) and/or LPS-stimulated cells; ^#^ *p* ≤ 0.05, not-digested coffee versus digested coffee.

**Table 1 nutrients-13-04368-t001:** UHPLC-MS parameters of the assayed analytes (*n* = 16).

Compound	Chemical	Adduct	RT	Measured	Theoretical	Accuracy
	Formula	Ion	(min)	Mass *(m*/*z)*	Mass *(m*/*z)*	(Δ mg/kg)
Quinic Acid	C_7_H_12_O_6_	[M−H]^−^	1.12	191.05531	191.05611	−4.19
5-CQA	C_16_H_18_O_9_	[M−H]^−^	3.18	353.0879	353.08780	0.03
4-CQA	C_16_H_18_O_9_	[M−H]^−^	3.19	353.08768	353.08780	−0.34
Caffeic Acid	C_9_H_8_O_4_	[M−H]^−^	3.20	179.03442	179.03498	−3.13
Caffeine	C_8_H_10_N_4_O_2_	[M+H]+	3.21	195.08757	195.08765	−0.41
3-CQA	C_16_H_18_O_9_	[M−H]^−^	3.22	353.08762	353.08780	−0.51
3-*p*CoQA	C_16_H_18_O_8_	[M−H]^−^	3.31	337.09232	337.09289	−1.69
5-*p*CoQA	C_16_H_18_O_8_	[M−H]^−^	3.32	337.0929	337.09289	0.03
3-FQA	C_17_H_20_O_9_	[M−H]^−^	3.39	367.10309	367.10346	−1.01
4+5-FQA	C_17_H_20_O_9_	[M−H]^−^	3.40	367.10303	367.10346	−1.17
4,5-CFQA	C_26_H_26_O_12_	[M−H]^−^	3.43	529.13495	529.13245	4.72
Ferulic Acid	C_10_H_10_O_4_	[M−H]^−^	3.46	193.05017	193.05063	−2.38
*p*-Coumaric acid	C_9_H_8_O_3_	[M−H]^−^	3.48	163.03934	163.04006	−4.42
3,4-diCQA	C_25_H_24_O_12_	[M−H]^−^	3.50	515.12103	515.11950	2.97
3,5-diCQA	C_25_H_24_O_12_	[M−H]^−^	3.53	515.11993	515.11950	0.83
3,4-FCQA	C_26_H_26_O_12_	[M−H]^−^	3.65	529.13247	529.13245	0.04

Abbreviations: CQA: caffeoylquinic acid; *p*CoQA: *p*-coumaroylquinic acid; FQA: feruloylquinic acids; diCQA: dicaffeoylquinic acid.

**Table 2 nutrients-13-04368-t002:** CGA (*n* = 11), phenolic acids (*n* = 4), and caffeine content in the assayed samples.

Compounds	Not-Digested Coffee Brew		Digested Coffee Brew	
	mg/g	SD	mg/g	SD
3-CQA	3.40 *	0.43	4.97 *	0.69
4-CQA	5.05 *	0.05	6.92 *	0.06
5-CQA	21.40 *	0.43	25.97 *	0.69
3-CoQA	6.54	0.51	6.27	0.28
5-CoQA	7.19	0.25	7.53	0.96
3-FQA	9.21 *	0.38	10.56 *	0.44
4+5-FQA	11.90 *	0.52	14.28 *	0.53
3,4-diCQA	1.70	0.04	1.88	0.06
3,5-diCQA	0.53 *	0.02	0.74 *	0.01
3-FCQA	1.17 *	0.05	1.41 *	0.01
4-CFQA	0.83 *	0.02	0.98 *	0.01
TOTAL CGA	68.92 *		81.50 *	
Caffeic acid	0.19 *	0.01	1.55 *	0.03
Quinic acid	1.69 *	0.20	2.79 *	0.02
Ferulic acid	2.71 *	0.08	4.12 *	0.07
*p*-coumaric acid	0.71 *	0.08	1.12 *	0.07
TOTAL phenolic	5.29 *		9.57 *	
Caffeine	19.96 *	0.07	17.68 *	0.27

Differences between groups were statistically analyzed with Tukey’s test; * *p*-value ≤ 0.05 was considered significant. Abbreviations: CQA: caffeoylquinic acid; *p*CoQA: *p*-coumaroylquinic acid; FQA: feruloylquinic acids; diCQA: dicaffeoylquinic acid.

**Table 3 nutrients-13-04368-t003:** TPC and antioxidant capacity evaluated by FRAP, DPPH, and ABTS of the samples.

Samples	FRAP		DPPH		ABTS		TPC	
	mmol TE/100 g	±SD	mmol TE/100 g	±SD	mmol TE/100 g	±SD	mg GAE/g	±SD
Coffee Brew	67.1 *	3.2	22.8 *	0.6	44.6 *	1.2	97.07 *	1.52
Coffee Brew Digested	72.5 *	0.7	51.4 *	0.3	89.2 *	0.6	106.64 *	2.01

Differences between groups were statistically analyzed with Tukey’s test; * *p*-value ≤ 0.05 was considered significant.

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
