# Peer review of "Antioxidant and Anti-Inflammatory Activity of Coffee Brew Evaluated after Simulated Gastrointestinal Digestion"

_nutrients, 2021, doi:10.3390/nu13124368_

Round 1
Reviewer 1 Report
I agree with the author´s reply to my further comment and thepotential addition of the data to the revised version.
Author Response
-The authors thank Reviewer 1 for evaluating our manuscript.
Reviewer 2 Report
The article "Antioxidant and Anti-inflammatory Activity of Coffee Brew Evaluated after Simulated Gastrointestinal Digestion" is an interesting paper that tried to assess information about the changes in the coffee's anti-inflammatory and antioxidant activity after simulated gastrointestinal digestion. As part of the work, the authors also presented the characterization of samples using high-resolution mass spectrometry analysis. I find this work written carefully. However, I would propose some modifications to improve the manuscript.
- The methodology for antioxidant in vitro spectrophotometric assays should be extended and supplemented with concentrations of tested substances, Trolox concentrations necessary for creating the curve required to conversion to the equivalent; it should also describe the blanks used for the experiment.
- As above, authors should complete the methodology of total polyphenolic contents determination.
- A part of the discussion on the properties of chlorogenic acid, such as the ability to lower LDL, seems unnecessary if we relate the results to parts of the intestine where absorption is very limited (HT-29).
- It would also be worthwhile to discuss better the increase in antioxidant activity and its positive effect on health, considering the part of the digestive tract under consideration.
- Not all abbreviations (ROS, LPS) are explained in the abstract. The abstract should be treated as an independent part of the work, and Authors should specify the abbreviations used in it.
- I suggest that the abbreviations used in the table should be explained below, even if the abbreviation has already been described in the text.
- C. arabica L. - should be used with a full two-part name.
- The Authors should indicate what kind of antioxidant activity is described by the in vitro antioxidants tests.
- n = 14 (above Table 1) is not consistent with the value of n = 16 mentioned in the table.
Author Response
Reviewer 2
The article "Antioxidant and Anti-inflammatory Activity of Coffee Brew Evaluated after Simulated Gastrointestinal Digestion" is an interesting paper that tried to assess information about the changes in the coffee's anti-inflammatory and antioxidant activity after simulated gastrointestinal digestion. As part of the work, the authors also presented the characterization of samples using high-resolution mass spectrometry analysis. I find this work written carefully. However, I would propose some modifications to improve the manuscript.
1) The methodology for antioxidant in vitro spectrophotometric assays should be extended and supplemented with concentrations of tested substances, Trolox concentrations necessary for creating the curve required to conversion to the equivalent; it should also describe the blanks used for the experiment. As above, authors should complete the methodology of total polyphenolic contents determination.
- As suggested by Reviewer 2, the authors added the missing information as “Results were calculated from the calibration curve prepared in triplicate at 6 concentration levels (5 - 200 µM of Trolox)” in the antioxidant experiments and as “Results were calculated from the calibration curves prepared in triplicate at 6 concentration levels (0.25 - 0.01 mg/mL of gallic acid)”.
2) A part of the discussion on the properties of chlorogenic acid, such as the ability to lower LDL, seems unnecessary if we relate the results to parts of the intestine where absorption is very limited (HT-29). It would also be worthwhile to discuss better the increase in antioxidant activity and its positive effect on health, considering the part of the digestive tract under consideration.
- As suggested by Reviewer 2, the authors removed the unnecessary part and added the missing information as “Moreover, similar data were obtained by Campos-Vega et al., who demonstrated that both antioxidant capacity and the colon bioaccessibility of polyphenols from spent coffee, rich in CGAs and melanoidins, were significantly higher after colonic fermentation compared to the non-digested samples.
3) Not all abbreviations (ROS, LPS) are explained in the abstract. The abstract should be treated as an independent part of the work, and Authors should specify the abbreviations used in it.
- As suggested by Reviewer 2, the authors added the missing information
4) I suggest that the abbreviations used in the table should be explained below, even if the abbreviation has already been described in the text.
- As suggested by Reviewer 2, the Authors added the missing information
5) C. arabica L. - should be used with a full two-part name.
- As suggested by Reviewer 2, the authors changed the terms "C. arabica L " as "Coffea arabica L.".
6) The Authors should indicate what kind of antioxidant activity is described by the in vitro antioxidants tests.
- As suggested by Reviewer 2, the authors added the missing information as “The antioxidant activity of the digested and not-digested coffee brew samples was assessed by using two different free radical scavenging activity methods including ABTS and DPPH tests and the ferric ion reducing antioxidant power assay”. namely FRAP.
7) n = 14 (above Table 1) is not consistent with the value of n = 16 mentioned in the table
- As suggested by Reviewer 2, the authors corrected the number in n=16.
-The authors thank Reviewer 2 for evaluating our manuscript.
Reviewer 3 Report
The authors have made a great effort in the presentation of the methods and results of this work.
I would invite the authors to consider the following:
- Inclusion of a positive/negative control on the MTT assay
- It might be helpful for the reader to explain why IL evaluation was not carried for 48 and 72h
- Lines 523-524 consider rephrasing
- The authors have reported correlations between the compound analysis, the TPC and the antioxidant activity as exhibited via the evaluations of ABTS,FRAP and DPPH. It would be interesting to include potential positive correlations assessed, between the basic antioxidant assays and the evaluations in the cells. Is there a link that might be useful for future research?
Other than that this is a useful manuscript to be used as a starting point for further investigations.
Author Response
The authors have made a great effort in the presentation of the methods and results of this work. I would invite the authors to consider the following:
1) Inclusion of a positive/negative control on the MTT assay
Regarding this aspect, the authors wish to underline that the effect of treatments on cell viability was calculated as fold-change relative to the negative control (untreated cells), arbitrarily set at 100%, and included in the graphs as point 0 mg/mL. Furthermore, MTT assay after the treatment with vehicle (blank control) was separately illustrated in Figure 1D to better underline the results, excluding cytotoxic effects mediated by the digestion fluid for 24h, 48h, and 72h.
2) It might be helpful for the reader to explain why IL evaluation was not carried for 48 and 72h
- In the current study the interleukin evaluation was carried out after 48h. Briefly, cells firstly were exposed to treatment with not-digested and digested coffee alone for 24h and then co-exposed for other 24h to LPS. The authors chose 48h of treatment because, at this time, H2DCF-DA results had shown more significant antioxidant properties of digested coffee with respect to not-digested coffee.
3) Lines 523-524 consider rephrasing
- As suggested by Reviewer 3, the authors changed the sentence from “In particular when HT-29 human colon cancer cells were treated for 48h with the assayed samples, the digested coffee highlighted more effective antioxidant activity than the not-digested ones, suggesting that the higher phytochemicals found in the samples after the digestion process could play an important role in managing ROS production supposedly through a wide array of biological mechanisms of action” to “In particular when HT-29 human colon cancer cells were treated for 48h with the assayed samples, the digested coffee highlighted more effective antioxidant activity than the not-digested ones, suggesting that the higher phytochemicals found in the samples after the digestion process could play an important role in managing ROS production.”
4) The authors have reported correlations between the compound analysis, the TPC and the antioxidant activity as exhibited via the evaluations of ABTS,FRAP and DPPH. It would be interesting to include potential positive correlations assessed, between the basic antioxidant assays and the evaluations in the cells. Is there a link that might be useful for future research?
- The authors express their gratitude to Reviewer 3 for his insightful suggestions. Furthermore, the authors inform Reviewer 3 that they are finishing a new manuscript in which they will expand on these points.
Other than that this is a useful manuscript to be used as a starting point for further investigations.
-The authors thank Reviewer 3 for evaluating our manuscript.
This manuscript is a resubmission of an earlier submission. The following is a list of the peer review reports and author responses from that submission.
Round 1
Reviewer 1 Report
Review of Manuscript ID: nutrients-1392551
Title: Antiproliferative and anti-inflammatory activity of coffee brew evaluated after simulated gastrointestinal digestion on HT-29 human colon cancer cells
Authors: Luigi Castaldo *, Marianna Toriello, Raffaele Sessa, Luana Izzo *, Sonia Lombardi, Alfonso Narváez, Alberto Ritieni, Michela Grosso.
General comment: the study reports the biological effect of plain coffee brew and after a simulated gastrointestinal digestion. The authors systematically analyze chemical composition of digested and undigested samples and show concentration of main components in each sample. Besides, they analyze and compare antioxidant capacity of both samples by three different methods and test both samples for their ROS-quenching capacity and anti-inflammatory potential. Although the objective of showing that the anticancer effect of coffee brews on colonic cancer increases after in vitro gastrointestinal digestion is sound and interesting, the study presents some major concerns regarding the model, experimental design and concentrations selected. Specific comments are detailed below:
Specific comments:
- The authors should consider that in the present experimental approach the coffee brew components are literally protecting and strengthening tumorigenic cells (HT29) instead of damaging or growth arresting them. Indeed, reference 8 by Mojica et al. justifies the potential anti-cancer capacity of coffee in colon cancer by its growth inhibitory activity of HT-29 cells. The same applies to reference 61, where chlorogenic acid inhibits growth rate of HT29 cells in culture. The selection of the model cell culture and its suitability for the study should be addressed and explained in introduction and/or discussion to prevent contradictory conclusions.
- The title is contradictory since both coffee brews show anti-proliferative effect at very high unrealistic doses that are cytotoxic for cultured HT29 (figure 1). These pharmacological concentrations are indeed discarded by the authors for the rest of the study.
- Line 212; references 44-45 regarding conditions of HT-29 cell culture refer to melanoma and leukemia cell lines; perhaps the authors should look for others more appropriate.
- Lines 318-335; text is out of standard format.
- Line 340; concentrations of 250-750 microg/mL that authors consider as low, should indeed be considered as supra-physiological or pharmacological. Extract concentrations over 100 microg/mL in cell culture are generally regarded as out of the physiological range. The authors should address this point.
- Figure 2, incomplete statistical analysis; data from different concentrations of non-digested and digested brews should be compared among them, not only to untreated controls. There is no reason to test different doses of the same compound/product if they are not compared to show a potential dose-dependent response.
- Line 361, incorrect statement; as stated above, the authors cannot state “ROS decreased in a dose-dependent manner after treatment with coffee as compared to untreated cells” until data from two different concentrations have been properly compared. In fact, it does not look like there is any significant difference between the two concentrations of both samples at any time-point. The authors should be aware that a different number of asterisks does not indicate a statistical significance.
- Figures 2 and 3, antioxidant and anti-inflammatory capacity; the authors should consider that the experimental design to show both capacities is rather different. Antioxidant capacity is assayed directly on cells in a basal non-stressed state whereas anti-inflammatory capacity is tested in a condition of LPS-induced damage. The authors should address this crucial difference and explain the reason why one capacity is analyzed in steady-state conditions whereas the other is evaluated under challenging conditions.
- Line 415; reference 54 should be quoted in the text as Perez-Burillo et al.
- Line 416; for antioxidant potential of coffee melanoidins in cell culture, the authors may also see: Goya, L., Delgado-Andrade, C., Rufián-Henares, J.A., Bravo, L. and Morales, F.J. 2007. Effect of coffee Melanoidin on human hepatoma HepG2 cells. Protection against oxidative stress induced by tertbutyl hydroperoxide. Mol. Nutr. Food Res. 51:536-545.
- Line 461; first author in reference 8 is Mojica, it should be referred as Mojica et al. There is no author named Benigno in that reference.
Author Response
Manuscript ID: nutrients-1392551
Type of manuscript: Article
Title: Antiproliferative and anti-inflammatory activity of coffee brew evaluated after simulated gastrointestinal digestion on HT-29 human colon cancer cells
Reviewer 1
General comment: the study reports the biological effect of plain coffee brew and after a simulated gastrointestinal digestion. The authors systematically analyze chemical composition of digested and undigested samples and show concentration of main components in each sample. Besides, they analyze and compare antioxidant capacity of both samples by three different methods and test both samples for their ROS-quenching capacity and anti-inflammatory potential. Although the objective of showing that the anticancer effect of coffee brews on colonic cancer increases after in vitro gastrointestinal digestion is sound and interesting, the study presents some major concerns regarding the model, experimental design and concentrations selected. Specific comments are detailed below:
Specific comments:
1) The authors should consider that in the present experimental approach the coffee brew components are literally protecting and strengthening tumorigenic cells (HT29) instead of damaging or growth arresting them. Indeed, reference 8 by Mojica et al. justifies the potential anti-cancer capacity of coffee in colon cancer by its growth inhibitory activity of HT-29 cells. The same applies to reference 61, where chlorogenic acid inhibits growth rate of HT29 cells in culture. The selection of the model cell culture and its suitability for the study should be addressed and explained in introduction and/or discussion to prevent contradictory conclusions.
- The human colon adenocarcinoma cell line HT29 is a common experimental model in studies focused on food digestion and bioavailability due to their ability to express characteristics of mature intestinal cells (please kindly find more details in Daniel Martínez-Maqueda et al, in Verhoeckx K, Cotter P, López-Expósito I, Kleiveland C, Lea T, Mackie A, Requena T, Swiatecka D, Wichers H, editors. The Impact of Food Bioactives on Health: in vitro and ex vivo models. Springer; 2015). For this reason these cells are also used to investigate protective and potential anticancer effects of food compounds, and their gastrointestinal digests (ref: Juan Antonio Pérez-Vega, Leticia Olivera-Castillo, José Ángel Gómez-Ruiz, Blanca Hernández-Ledesma, Release of multifunctional peptides by gastrointestinal digestion of sea cucumber (Isostichopus badionotus), Journal of Functional Foods, Volume 5, Issue 2, 2013, Pages 869-877, ISSN 1756-4646, doi: 10.1016/j.jff.2013.01.036., Mojica BE, Fong LE, Biju D, Muharram A, Davis IM, Vela KO, Rios D, Osorio-Camacena E, Kaur B, Rojas SM, Forester SC. The Impact of the Roast Levels of Coffee Extracts on their Potential Anticancer Activities. J Food Sci. 2018 Apr;83(4):1125-1130. doi: 10.1111/1750-3841.14102. Epub 2018 Mar 25. PMID: 29577313, Shi J, Shan S, Li H, Song G, Li Z. Anti-inflammatory effects of millet bran derived-bound polyphenols in LPS-induced HT-29 cell via ROS/miR-149/Akt/NF-κB signaling pathway. Oncotarget. 2017 Aug 12;8(43):74582-74594. doi: 10.18632/oncotarget.20216. PMID: 29088809; PMCID: PMC5650364). Based on these observations, the authors believe that HT29 cells represent a suitable model for this study since the antioxidant and anti-inflammatory effects detected in these cells are to be considered as protective against transformation of normal intestinal epithelial cells into cancer cells without any implication for anti-cancer therapies. A sentence addressing this aspect has been added in the the following paragraph, introduction section: “Nowadays, the use of HT-29 human colon cancer cells to study at the cellular level the effect of coffee on colon cancer cell growth has been used in several experiments. Recently, spent coffee grounds have been reported to induce HT-29 cell apoptosis by reducing 8-iso-prostaglandin F2α and catalase after simulated colonic digestion [33]. Moreover, 5- caffeoylquinic acid (5-CQA) from coffee brews has been found to reduce ROS production in HT-29 human colon cell model [34]. Furthermore, the level of roasting was reported to affect cell viability and the results showed that lighter roasted coffee reduced HT-29 cell growth more than darker roasted samples [10].
2) The title is contradictory since both coffee brews show anti-proliferative effect at very high unrealistic doses that are cytotoxic for cultured HT29 (figure 1). These pharmacological concentrations are indeed discarded by the authors for the rest of the study.
- As suggested by reviewer 1, the authors changed the title in “Antioxidant and Anti-inflammatory Activity of Coffee Brew after Simulated Gastrointestinal Digestion”
3) Line 212; references 44-45 regarding conditions of HT-29 cell culture refer to melanoma and leukemia cell lines; perhaps the authors should look for others more appropriate.
- As suggested by reviewer 1, these references were replaced with a more appropriate one.
4) Lines 318-335; text is out of standard format.
- As suggested by reviewer 1, the authors fixed the text out of standard.
5) Line 340; concentrations of 250-750 microg/mL that authors consider as low, should indeed be considered as supra-physiological or pharmacological. Extract concentrations over 100 microg/mL in cell culture are generally regarded as out of the physiological range. The authors should address this point.
- Thank you for this comment. In the revised manuscript the authors clarified that these concentrations were chosen according to what reported in other similar studies (Mojica BE, Fong LE, Biju D, Muharram A, Davis IM, Vela KO, Rios D, Osorio-Camacena E, Kaur B, Rojas SM, Forester SC. The Impact of the Roast Levels of Coffee Extracts on their Potential Anticancer Activities. J Food Sci. 2018 Apr;83(4):1125-1130. doi: 10.1111/1750-3841.14102. Epub 2018 Mar 25. PMID: 29577313).
6) Figure 2, incomplete statistical analysis; data from different concentrations of non-digested and digested brews should be compared among them, not only to untreated controls. There is no reason to test different doses of the same compound/product if they are not compared to show a potential dose-dependent response.
- Thank you for this comment. As suggested by reviewer 1, the authors completed statistical analysis between not-digested and digested brews.
7) Line 361, incorrect statement; as stated above, the authors cannot state “ROS decreased in a dose-dependent manner after treatment with coffee as compared to untreated cells” until data from two different concentrations have been properly compared. In fact, it does not look like there is any significant difference between the two concentrations of both samples at any time-point. The authors should be aware that a different number of asterisks does not indicate a statistical significance.
- As suggested by reviewer 1, the authors revised the text in the manuscript.
7) Figures 2 and 3, antioxidant and anti-inflammatory capacity; the authors should consider that the experimental design to show both capacities is rather different. Antioxidant capacity is assayed directly on cells in a basal non-stressed state whereas anti-inflammatory capacity is tested in a condition of LPS-induced damage. The authors should address this crucial difference and explain the reason why one capacity is analyzed in steady-state conditions whereas the other is evaluated under challenging conditions.
-The Authors are thankful to Reviewer 1 for this comment. Following this suggestion, additional experiments were performed to detect ROS levels in cells treated with LPS with or without the analyzed coffee samples. Results are shown in Fig. 4.
8) Line 415; reference 54 should be quoted in the text as Perez-Burillo et al.
- As suggested by reviewer 1, the authors changed reference 54 in the text as Perez-Burillo.
9) Line 416; for antioxidant potential of coffee melanoidins in cell culture, the authors may also see: Goya, L., Delgado-Andrade, C., Rufián-Henares, J.A., Bravo, L. and Morales, F.J. 2007. Effect of coffee Melanoidin on human hepatoma HepG2 cells. Protection against oxidative stress induced by tertbutyl hydroperoxide. Mol. Nutr. Food Res. 51:536-545.
-As suggested by reviewer 1, the authors added the reference.
10) Line 461; first author in reference 8 is Mojica, it should be referred as Mojica et al. There is no author named Benigno in that reference.
- As suggested by reviewer 1, the authors corrected the author’s name in the reference section
The authors thank Reviewer 1 for evaluating the manuscript.
Reviewer 2 Report
The paper entitled “Antiproliferative and Anti-inflammatory Activity of Coffee Brew Evaluated after Simulated Gastrointestinal Digestion on HT-29 Human Colon Cancer Cells” by Luigi Castaldo et al. reports an interesting study. The paper provides some novel information about the potential of coffee and their metabolites as chlorogenic acids to inhibit intracellular reactive oxygen species and pro-inflammatory activity on human colon cells.
Nowadays the topic is relevant. This paper is well-written and presents an interesting and carefully designed research. The work is very detailed and sometimes the authors repeat the same concepts along the paper which is hard to read. Some minor issues should be resolved.
Title
It is too long, considering to change it
Introduction
Introduction section is too extensive. Please try to reduce it.
Line 41: please incorporate more recent references as “Sánchez-Quesada C, Romanos-Nanclares A, Navarro AM, Gea A, Cervantes S, Martínez-González MÁ, Toledo E. Coffee consumption and breast cancer risk in the SUN project. Eur J Nutr. 2020 Dec;59(8):3461-3471. doi: 10.1007/s00394-020-02180-w. Epub 2020 Jan 18. PMID: 31955220.”
Line 53-54: According to Dietary Guidelines, a moderate coffee consumption is defined as 3-5 cups per day. “moderate coffee consumption (≥ 5 cups per day)” should be change by “high coffee consumption”
Line 75-76: The sentence “several studies” should be accompanied by more references.
Line 92-93: The sentence “Therefore, this study aimed to assess the changes occurring in polyphenol coffee composition after simulated gastrointestinal digestion (GiD)”, should be deleted. This study aimed to analysed the effects of polyphenol coffee after GiD in colon human cells. The other goal was evaluated by authors in the paper entitled “Colon Bioaccessibility under In Vitro Gastrointestinal Digestion of Different Coffee Brews Chemically Profiled through UHPLC-Q-Orbitrap HRMS”.
Methods
Line 209: Have you performed the trials on normal human colonic cell lines? Because of non tumoral cells has a low range of ROS levels than tumoral cells. Furthermore, some antioxidants enzymes as catalase, superoxide dismutase are absent in these cancer cells. A non tumoral model will emphasise the results obtained in this work.
Results
Table 3: please incorporate the p value
Line 337: “Effect of not-Digested or Digested Coffee on Cell Viability in HT-29 Cells”. Review statistical analysis
Figure 1: consider using another colour for digested coffee
Figures: Error bars are missing in all the controls
Discussion
Line 441. Please change “that these tests were“ for “that these test could be”.
Conclusion
The first and second paragraphs of the conclusions have already been published in another article by the authors (“In conclusion, in this study a high-resolution Orbitrap mass spectrometry analysis allowed to obtain a comprehensive chemical characterization of the major bioactive molecules including caffeine, CGA (n=9), and phenolic acids (n=4) in coffee brews samples, before and after in vitro GiD. Data clearly indicate that the GiD process affected the monitored bioactive compounds, in particular, the levels found in coffee digested samples were higher than the non-digested ones, suggesting that the digestive process was able to release dietary polyphenols incorporated into the complex structure of the melanoidins”).
Lines 487-491. These paragraphs are more for discussion section that for the final conclusion. Indeed, you has to argument these facts with references.
Author Response
Manuscript ID: nutrients-1392551
Type of manuscript: Article
Title: Antiproliferative and anti-inflammatory activity of coffee brew evaluated after simulated gastrointestinal digestion on HT-29 human colon cancer cells
Reviewer 2
Title
1) It is too long, considering to change it
- As suggested by reviewer 1, the authors changed the title of the manuscript in “Antioxidant and Anti-inflammatory Activity of Coffee Brew after Simulated Gastrointestinal Digestion”
Introduction
2) Introduction section is too extensive. Please try to reduce it.
- As suggested by reviewer 2, the authors tried to reduce the introduction section, however, the authors had to add some additional information as requested by other Reviewers.
3) Line 41: please incorporate more recent references as “Sánchez-Quesada C, Romanos-Nanclares A, Navarro AM, Gea A, Cervantes S, Martínez-González MÁ, Toledo E. Coffee consumption and breast cancer risk in the SUN project. Eur J Nutr. 2020 Dec;59(8):3461-3471. doi: 10.1007/s00394-020-02180-w. Epub 2020 Jan 18. PMID: 31955220.”
-As suggested by reviewer 2, the authors added the reference.
4) Line 53-54: According to Dietary Guidelines, a moderate coffee consumption is defined as 3-5 cups per day. “moderate coffee consumption (≥ 5 cups per day)” should be change by “high coffee consumption”
As rightly suggested by reviewer 2, the authors changed the term "moderate" as "high". Moreover, the authors added further details for better understanding as “According to Dietary Guidelines for Americans 2015-2020 as up to five cups of coffee per day or providing up to 400 mg/day of caffeine”
5) Line 75-76: The sentence “several studies” should be accompanied by more references.
As suggested by reviewer 2, the authors added the references
6) Line 92-93: The sentence “Therefore, this study aimed to assess the changes occurring in polyphenol coffee composition after simulated gastrointestinal digestion (GiD)”, should be deleted. This study aimed to analysed the effects of polyphenol coffee after GiD in colon human cells. The other goal was evaluated by authors in the paper entitled “Colon Bioaccessibility under In Vitro Gastrointestinal Digestion of Different Coffee Brews Chemically Profiled through UHPLC-Q-Orbitrap HRMS”.
As suggested by reviewer 2, the authors removed the sentence in the text
Methods
6) Line 209: Have you performed the trials on normal human colonic cell lines? Because of non tumoral cells has a low range of ROS levels than tumoral cells. Furthermore, some antioxidants enzymes as catalase, superoxide dismutase are absent in these cancer cells. A non tumoral model will emphasise the results obtained in this work.
From a general point of view, authors agree with Reviewer that normal colon cell lines might represent a more physiologically relevant setting for these studies. However, given their ability to multiply indefinitely, all immortalized cell lines accumulate mutations over time in culture and may partially lose the phenotypic features that are found in the parental cells. Therefore, only primary cell cultures from tissue biopsies could resemble a normal phenotype. On the other hand, it is to be considered that biopsies from normal tissues are generally extremely difficult to be obtained as well as that cell lines are well characterized and are more homogeneous compared to primary cultures. Consequently, despite several limitations generally shown by human colon carcinoma cell lines, HT29 cells represent a valuable accessible and easily usable model due to their similarities with enterocytes of the small intestine to study several aspects related with food digestion and bioavailability of food compounds. Therefore, based on these considerations, HT29 cells are to be considered a suitable experimental model for this study.
Results
7) Table 3: please incorporate the p value
As suggested by reviewer 2, the authors added the missing information
8) Line 337: “Effect of not-Digested or Digested Coffee on Cell Viability in HT-29 Cells”. Review statistical analysis
- As suggested by reviewer 2, the authors revised the manuscript and added details regarding statistical significance in cell viability variations between not-digested or digested coffee samples.
9) Figure 1: consider using another colour for digested coffee
- As suggested by reviewer 2, the authors changed the colour for digested coffee that is now reported in red.
10) Figures: Error bars are missing in all the controls
- As suggested by reviewer 2, the authors added the missing information
Discussion
11) Line 441. Please change “that these tests were“ for “that these test could be”.
As suggested by reviewer 2, the authors changed the sentence
Conclusion
12) The first and second paragraphs of the conclusions have already been published in another article by the authors (“In conclusion, in this study a high-resolution Orbitrap mass spectrometry analysis allowed to obtain a comprehensive chemical characterization of the major bioactive molecules including caffeine, CGA (n=9), and phenolic acids (n=4) in coffee brews samples, before and after in vitro GiD. Data clearly indicate that the GiD process affected the monitored bioactive compounds, in particular, the levels found in coffee digested samples were higher than the non-digested ones, suggesting that the digestive process was able to release dietary polyphenols incorporated into the complex structure of the melanoidins”).
As suggested by reviewer 2, the authors removed the first and second paragraphs in the conclusion section.
13) Lines 487-491. These paragraphs are more for discussion section that for the final conclusion. Indeed, you has to argument these facts with references.
As suggested by reviewer 2, the authors added the missing information in the introduction section to argue these facts as “Nowadays, the use of HT-29 human colon cancer cells to study at the cellular level the effect of coffee on colon cancer cell growth has been used in several experiments. Recently, spent coffee grounds have been reported to induce HT-29 cell apoptosis by reducing 8-iso-prostaglandin F2α and catalase after simulated colonic digestion [33]. Moreover, 5-caffeoylquinic acid (5-CQA) from coffee brews has been found to reduce ROS production in HT-29 human colon cell model [34]. Furthermore, the level of roasting was reported to affect cell viability and the results showed that lighter roasted coffee reduced HT-29 cell growth more than darker roasted samples [10]”.
The authors thank Reviewer 2 for evaluating the manuscript.
Reviewer 3 Report
Nutrients 1392551
“Antiproliferative and Anti-inflammatory Activity of Coffee Brew Evaluated after Simulated Gastrointestinal Digestion on HT-29 Human Colon Cancer Cells”
The authors investigated the content of polyphenols and the antioxidant activity of in vitro digested coffee brew in comparison to not-digested coffee brew. In addition, cell culture experiments were performed to examine cytotoxic effects, intracellular ROS reduction and expression of pro- and anti-inflammatory proteins.
Though the experimental work has been well done and the manuscript is well organized the experimental setting and results presented in the manuscript lack a little novelty. The MTT assay alone is not suitable to derive antiproliferative effects of the used samples. Furthermore, the experimental settings lack appropriate controls as for example there is no positive control for ROS-formation used in the DCFH-DA-Assay. Most importantly, no vehicle or blank control resulting from the in vitro digestion experiments were used for further experiments. Such controls are essential to investigate the effects in cell culture experiments. Furthermore, the discussion is not profound enough.
The following remarks should be suited to improve the manuscript.
- Title: The authors did not show antiproliferative effects of their samples. Therefore, the title should be reworded.
- Abstract
- Lines 29-30: The conclusion should be reworded to a more cautious message because the authors not really investigated anti-cancer effects of their samples
- Introduction
- Lines 52 and 54: Define high and moderate coffee consumption more clearly.
- The authors should mention the formation of potential harmful contaminants such as acrylamide
- Line 83: check the grammar of: CGAs reach intact the lower intestine
- Line 84: microflora or microbiota?
- Line 95: Explain the rationale for using colon cancer cells to study antioxidant and anti-inflammatory properties
- The authors should explain the novelty of the work
- Material and Methods
- Line 120: The correct city would be: Emmerich am Rhein
- Line 134: Please correct: H2O
- Line 160: The authors should explain the basis of calculation of the amounts of sample used for in vitro digestion (why 5 ml of coffee brew?). It should be stated here, or in the discussion section, how these amount of coffee does reflect the in vivo situation of coffee consumption
- Line 173 and 174: what is the rationale to use enzymes to mimic the colonic situation instead of using feces samples as bacterial source and anaerobe atmospheric conditions to better mimic the in vivo situation?
- Appropriate controls such as a blank digestion control should be included in the experimental setting of in vitro digestion
- Line 222: please include information why these concentrations were chosen
- Line 231: please include information how formazan crystals were dissolved
- Lines 274-275: the indication of significance is unconventional since p < 0.01 and p < 0.001 are usually labelled with ** and ***, respectively
- Results
- All Tables: Please include information about abbreviation in the legends
- All Tables: Please include significances
- Lines 304-307: Please check results regarding CQAs.
- Lines 309-311: Please check results regarding FQAs
- Line 317: 1.41 mg/g (see Table 2)
- Line 320: please include significant differences also in the tables
- Line 343: The authors state, that cell viability was reduced in a time and dose-dependent manner, this is not quite the fact, please check
- Figure 1: explain how results were calculated
- Figure 2: explain how results were calculated, is it a fold-change? This assay lacks a positive control! Please include a positive control such as TBH or H2O2
- Discussion
- The authors should clearly state what is the novelty of the work
- The authors should discuss the suitability of the used experimental setting for the in vitro digestion, mainly the use of the protease mixtures…Is there any information in the literature how good this procedure would fit the in vivo situation?
- The authors should clarify the used amounts of coffee brew used for the in vitro digestion procedure and how these amounts are transferable to the in vivo situation
- A more profound discussion is needed regarding the concentrations of polyphenols found in the coffee brew samples and the correlating amounts used for cell culture experiments. Are these concentrations comparable to other studies?
- Is it true that the used samples exert activity against oxidative stress? The authors investigated the reduction of basal ROS levels. They did not investigate a reduction of excessive ROS formation in HT29-cells. Therefore, the conclusions drawn from this results are too speculative.
- What effects might be responsible for ROS reduction. Explain in more detail
- Lines 461-465: The authors should explain in more detail, what might be reason for the discrepancies of the study of Benigno et al. and their own, if there are any? What concentrations or amounts were used and were these comparable to the presented study? Furthermore, the statements are contradictory, please check
- Line 472: the authors investigates the protein expression of the cytokines in whole cell lysates not the secretion?
- Conclusion
- Line 489: The authors did not show anticancer activity or effects. This statement has to be reworded.
Author Response
Manuscript ID: nutrients-1392551
Type of manuscript: Article
Title: Antiproliferative and anti-inflammatory activity of coffee brew evaluated after simulated gastrointestinal digestion on HT-29 human colon cancer cells
Reviewer 3
The authors investigated the content of polyphenols and the antioxidant activity of in vitro digested coffee brew in comparison to not-digested coffee brew. In addition, cell culture experiments were performed to examine cytotoxic effects, intracellular ROS reduction and expression of pro- and anti-inflammatory proteins.
Though the experimental work has been well done and the manuscript is well organized the experimental setting and results presented in the manuscript lack a little novelty. The MTT assay alone is not suitable to derive antiproliferative effects of the used samples. Furthermore, the experimental settings lack appropriate controls as for example there is no positive control for ROS-formation used in the DCFH-DA-Assay. Most importantly, no vehicle or blank control resulting from the in vitro digestion experiments were used for further experiments. Such controls are essential to investigate the effects in cell culture experiments. Furthermore, the discussion is not profound enough.
The following remarks should be suited to improve the manuscript.
1) Title: The authors did not show antiproliferative effects of their samples. Therefore, the title should be reworded.
- The authors took this point and accordingly changed the title in “Antioxidant and Anti-inflammatory Activity of Coffee Brew Evaluated after Simulated Gastrointestinal Digestion”
Abstract
2) Lines 29-30: The conclusion should be reworded to a more cautious message because the authors not really investigated anti-cancer effects of their samples
As suggested by reviewer 3, the authors reworded the conclusion as “Overall, our findings suggest that coffee may exert antioxidant and anti-inflammatory properties, and the digested process may be able to release compounds with increased bioactivity.”
Introduction
3) Lines 52 and 54: Define high and moderate coffee consumption more clearly.
As suggested by reviewer 3, the authors added the missing information as “According to Dietary Guidelines for Americans 2015-2020 as up to five cups of coffee per day or providing up to 400 mg/day of caffeine”
4) The authors should mention the formation of potential harmful contaminants such as acrylamide
As suggested by reviewer 3, the authors added the missing information as “Although previous studies have reported adverse effects related to coffee consumption [2], mainly due to the presence of potential harmful contaminants such as acrylamide [3], a growing amount of scientific data support the potential benefits of regular coffee intake for human health”
5) Line 83: check the grammar of: CGAs reach intact the lower intestine
As suggested by reviewer 3, the authors checked the grammar as “CGAs reach the lower intestine in their intact form”
6) Line 84: microflora or microbiota?
As suggested by reviewer 3, the authors changed the sentence as “where they are metabolized by the gut microbiota”
7) Line 95: Explain the rationale for using colon cancer cells to study antioxidant and anti-inflammatory properties.
As suggested by reviewer 3, the authors added missing information in the text as “Nowadays, the use of HT-29 human colon cancer cells to study at the cellular level the effect of coffee on colon cancer cell growth has been used in several experiments. Recently, spent coffee grounds have been reported to induce HT-29 cell apoptosis by reducing 8-iso-prostaglandin F2α and catalase after simulated colonic digestion [33]. Moreover, 5- caffeoylquinic acid (5-CQA) from coffee brews has been found to reduce ROS production in HT-29 human colon cell model [34]. Furthermore, the level of roasting was reported to affect cell viability and the results showed that lighter roasted coffee reduced HT-29 cell growth more than darker roasted samples [10]”.
8) The authors should explain the novelty of the work
As suggested by reviewer 3, the authors explain the novelty of the work in the text of the manuscript as “Although the antioxidant and anti-inflammatory activities of coffee polyphenols have been widely studied, more in-depth knowledge is needed to clarify the effect of the gastrointestinal process on the coffee composition and their bioactivities in protecting against colon cancer. Therefore, this study aimed to assess the changes occurring in polyphenol composition and the relative differences in antioxidant and anti-inflammatory properties of coffee brews after simulated gastrointestinal digestion (GiD) through the HT-29 human colon cancer cell model.”
Material and Methods
9) Line 120: The correct city would be: Emmerich am Rhein
As suggested by reviewer 3, the authors corrected the name of the city.
10) Line 134: Please correct: H2O
As suggested by reviewer 3, the authors corrected the chemical formula.
11) Line 160: The authors should explain the basis of calculation of the amounts of sample used for in vitro digestion (why 5 ml of coffee brew?). It should be stated here, or in the discussion section, how these amount of coffee does reflect the in vivo situation of coffee consumption.
The protocol used to simulate human gastrointestinal digestion in the present study was recently developed in the COST action INFOGEST network. The above protocol is recognized as the most eligible method for a comparison of results among different labs using similar and close conditions. In the protocol proposed by INFOGEST the amount of sample is 5 g of solid or 5 mL of liquid food, for these reasons the authors chose the amounts of sample used for in vitro digestion.
12) Line 173 and 174: what is the rationale to use enzymes to mimic the colonic situation instead of using feces samples as bacterial source and anaerobe atmospheric conditions to better mimic the in vivo situation?
As suggested by reviewer 3, it is well-established that techniques using fecal inoculum are the most suitable method for the study of colonic digestion, mimicking the activity of the microbiota. However, previous studies have proposed the use of Pronase E and Viscozyme L to reproduce the activity of the gut microbiota as an effective alternative to the use of fecal inoculum
13) Appropriate controls such as a blank digestion control should be included in the experimental setting of in vitro digestion.
As suggested by reviewer 3, the authors added in the supplementary materials the TPC and the antioxidant capacity evaluated by FRAP, DPPH and ABTS of the blank control resulting from the in vitro digestion experiments.
14) Line 222: please include information why these concentrations were chosen
- The authors took this point and included the required information in the manuscript.
15) Line 231: please include information how formazan crystals were dissolved
- As suggested by Reviewer 3, in the revised manuscript the authors included the missing information.
16) Lines 274-275: the indication of significance is unconventional since p < 0.01 and p < 0.001 are usually labelled with ** and ***, respectively
- As suggested by reviewer 3, the authors checked statistical analysis adding conventional p-value information for p ≤ 0.05, p ≤ 0.01 and p ≤ 0.001 labelled with *, ** and ***, respectively.
Results
17) All Tables: Please include information about abbreviation in the legends
As suggested by reviewer 3, the authors added the missing information
18) All Tables: Please include significances
As suggested by reviewer 3, the authors added the missing information
19) Lines 304-307: Please check results regarding CQAs.
As suggested by reviewer 3, the authors corrected the results
20) Lines 309-311: Please check results regarding FQAs
As suggested by reviewer 3, the authors corrected the results
21) Line 317: 1.41 mg/g (see Table 2)
As suggested by reviewer 3, the authors corrected the results
22) Line 320: please include significant differences also in the tables
As suggested by reviewer 3, the authors added the missed information
23) Line 343: The authors state, that cell viability was reduced in a time and dose-dependent manner, this is not quite the fact, please check
- The Authors took this point and modified the text in the manuscript
24) Figure 1: explain how results were calculated
- As suggested by reviewer 3, in the revised manuscript the authors explained that results had been calculated as a fold-change related to the untreated control arbitrarily set at 100%.
25) Figure 2: explain how results were calculated, is it a fold-change? This assay lacks a positive control! Please include a positive control such as TBH or H2O2
- The Authors are grateful to the reviewer for this comment. Figure 2 now includes H2O2 as positive control and describes how results were calculated.
Discussion
26) The authors should clearly state what is the novelty of the work
As suggested by reviewer 3, the authors added the missing information as “The present study aimed to provide useful information regarding the bioactivities of coffee brews after simulated GiD. Despite the many scientific works present in the literature concerning the beneficial activities of coffee polyphenols in protecting against colon cancer, the study of bioactivities of compounds released after in vitro digestion process has been barely investigated to date. Hence, the main goal of this work was to investigate the antioxidant and anti-inflammatory activities of coffee brews after GiD through the HT-29 human colon cancer cell model.”
27) The authors should discuss the suitability of the used experimental setting for the in vitro digestion, mainly the use of the protease mixtures…Is there any information in the literature how good this procedure would fit the in vivo situation?
As suggested by reviewer 3, the authors added the missing information as “Although the use of the fecal inoculum is recognized as the most suitable method to simulate in vitro colonic digestion, an ever-expanding amount of scientific works propose as an effective alternative to reproduce the intestinal fermentation the use of a mix of bacterial enzymes, such as Viscozyme L and Pronase E.”
28) The authors should clarify the used amounts of coffee brew used for the in vitro digestion procedure and how these amounts are transferable to the in vivo situation
29) A more profound discussion is needed regarding the concentrations of polyphenols found in the coffee brew samples and the correlating amounts used for cell culture experiments. Are these concentrations comparable to other studies?
As suggested by reviewer 3, the authors added the missing information.as “In fact, Mojica et al., [10] compared the impact of the different coffee extracts on the growth inhibitory activity of HT-29 cells. The cells were treated with 1 to 50x dilutions of the coffee stock solutions and the coffee samples were tested without a previous simulated GiD. The authors reported that the coffee extract that had the greatest content of polyphenolic compounds was able to reduce the cell growth more than samples with the lowest amount of polyphenols”.
30) Is it true that the used samples exert activity against oxidative stress? The authors investigated the reduction of basal ROS levels. They did not investigate a reduction of excessive ROS formation in HT29-cells. Therefore, the conclusions drawn from this result are too speculative.
-The authors are thankful to Reviewer for this insightful comment. Following this suggestion, additional experiments were performed to detect ROS levels in cells treated with LPS with or without the analyzed coffee samples. Results are shown in Fig. 4.
31) What effects might be responsible for ROS reduction. Explain in more detail
Following this comment, a sentence was added regarding the ROS scavenger activity of the coffee extracts.
32) Lines 461-465: The authors should explain in more detail, what might be reason for the discrepancies of the study of Benigno et al. and their own, if there are any? What concentrations or amounts were used and were these comparable to the presented study? Furthermore, the statements are contradictory, please check
As suggested by reviewer 3, the authors added the missing information as “Cells were treated with either 1 to 50× dilutions of the coffee stock solutions and the coffee samples were assayed without GiD”. Moreover, the authors clarify this section.
33) Line 472: the authors investigate the protein expression of the cytokines in whole cell lysates not the secretion?
- The authors investigated the protein expression of the cytokines in whole cell lysate according to literature data (Brinkhoff A, Sieberichs A, Engler H, Dolff S, Benson S, Korth J, Schedlowski M, Kribben A, Witzke O, Wilde B. Pro-Inflammatory Th1 and Th17 Cells Are Suppressed During Human Experimental Endotoxemia Whereas Anti-Inflammatory IL-10 Producing T-Cells Are Unaffected. Front Immunol. 2018 May 18;9:1133. doi: 10.3389/fimmu.2018.01133. PMID: 29868038; PMCID: PMC5968108. Lee WS, Shin JS, Jang DS, Lee KT. Cnidilide, an alkylphthalide isolated from the roots of Cnidium officinale, suppresses LPS-induced NO, PGE2, IL-1β, IL-6 and TNF-α production by AP-1 and NF-κB inactivation in RAW 264.7 macrophages. Int Immunopharmacol. 2016 Nov;40:146-155. doi: 10.1016/j.intimp.2016.08.021. Epub 2016 Sep 1. PMID: 27591413; Shi J, Shan S, Li H, Song G, Li Z. Anti-inflammatory effects of millet bran derived-bound polyphenols in LPS-induced HT-29 cell via ROS/miR-149/Akt/NF-κB signaling pathway. Oncotarget. 2017 Aug 12;8(43):74582-74594. doi: 10.18632/oncotarget.20216. PMID: 29088809; PMCID: PMC5650364). The text has been modified to specify this aspect.
Conclusion
34) Line 489: The authors did not show anticancer activity or effects. This statement has to be reworded.
- As suggested by reviewer 3, the authors reworded this sentence.
The authors thank Reviewer 3 for evaluating the manuscript.
Round 2
Reviewer 1 Report
The authors have conveniently addressed most of my comments and queries, but there is still a point that needs to be considered. Following my comment regarding the differential testing of antioxidant and anti-inflammatory potential in basal and challenging conditions respectively, the authors have assayed and showed the effect of hydrogen peroxide on HT29 cells as positive control for oxidative stressed cells in figure 4; however, this result of positive control alone is useless unless the antioxidant capacity of the coffee samples is tested under these stressful conditions. ROS concentrations should be tested in the presence of coffee samples and hydrogen peroxide (as in the anti-inflammatory test with coffee samples and LPS) and the results shown in figure 4.
Author Response
Manuscript ID: nutrients-1392551
Type of manuscript: Article
Title: Antiproliferative and anti-inflammatory activity of coffee brew evaluated after simulated gastrointestinal digestion on HT-29 human colon cancer cells
Reviewer 1
1) The authors have conveniently addressed most of my comments and queries, but there is still a point that needs to be considered. Following my comment regarding the differential testing of antioxidant and anti-inflammatory potential in basal and challenging conditions respectively, the authors have assayed and showed the effect of hydrogen peroxide on HT29 cells as positive control for oxidative stressed cells in figure 4; however, this result of positive control alone is useless unless the antioxidant capacity of the coffee samples is tested under these stressful conditions. ROS concentrations should be tested in the presence of coffee samples and hydrogen peroxide (as in the anti-inflammatory test with coffee samples and LPS) and the results shown in figure 4.
1) Regarding this aspect, the authors wish to underline that LPS treatment promotes intestinal inflammation by inducing intracellular ROS production and oxidative injury in several cell models, including HT-29 cells [53-55]. Therefore, as also shown in Figure 4, LPS treatment induced increased intracellular ROS levels similarly to H2O2, thus acting as a challenging condition for ROS production. Accordingly, this issue has been addressed in paragraphs 2.10 and 3.6.
The authors thank Reviewer 1 for evaluating the manuscript.
Reviewer 3 Report
The authors improved the quality of their manuscript. But, a vehicle or blank control resulting from the in vitro digestion procedure is still missing in the cell culture experiments. Such a control is mandatory and essential to verify the real effects mediated by the digested coffee and to exclude effects mediated by the digestion fluid.
Author Response
Manuscript ID: nutrients-1392551
Type of manuscript: Article
Title: Antiproliferative and anti-inflammatory activity of coffee brew evaluated after simulated gastrointestinal digestion on HT-29 human colon cancer cells
Reviewer 3
1) The authors improved the quality of their manuscript. But, a vehicle or blank control resulting from the in vitro digestion procedure is still missing in the cell culture experiments. Such a control is mandatory and essential to verify the real effects mediated by the digested coffee and to exclude effects mediated by the digestion fluid.
1) As suggested by Reviewer 3, the authors added in the supplementary materials the effects of the blank control resulting from the in vitro digestion experiments on cell viability and in intracellular ROS level evaluated in HT-29 cells (Figure S1 and S2).
The authors thank Reviewer 3 for evaluating the manuscript.